# Independent and combined associations of sleep duration and sleep quality with common physical and mental disorders: Results from a multi-ethnic population-based study

**Lee Seng Esmond Seow**[1]*, Xiao Wei Tan[1], Siow Ann Chong[1], Janhavi Ajit Vaingankar[1], Edimansyah Abdin[1], Saleha Shafie[1], Boon Yiang Chua[1], Derrick Heng[2], Mythily Subramaniam[1]

1 Research Division, Institute of Mental Health, Singapore, Singapore, 2 Ministry of Health of Singapore, Singapore, Singapore

* esmond_LS_SEOW@imh.com.sg

**Data Availability Statement:** All individual data from this study resides with Office of Research,

## Abstract

Sleep duration and sleep quality are often linked to increased risk of mortality and morbidity. However, national representative data on both sleep duration and sleep quality and their relationship with chronic health problems are rarely available from the same source. This current study aimed to examine the independent and combined associations of sleep duration and sleep quality with physical and mental disorders, using data from the Singapore Mental Health Study 2016. 6,126 residents aged ≥18years participated in this epidemiological, cross-sectional survey. Sleep measures were assessed using the Pittsburg Sleep Quality Index while lifetime or 12-month medical and psychiatric diagnoses were established using the Composite International Diagnostic Interview 3.0. Both short sleep (<6hrs compared to 7-8hrs) and poor sleep were found to be independently associated with chronic pain, obsessive compulsive disorder and any mental disorder while poor sleep was additionally associated with major depressive disorder, bipolar disorder, generalized anxiety disorder and any physical disorder, when adjusted for confounders. Poor sleep combined with short sleep (≤6hrs/day vs 7-8hrs/day) was associated with the highest number of comorbidities among other sleep combinations. Sleep duration and sleep quality, when adjusted for each other, remained independently associated with both physical and mental disorders. Affective disorders may be more closely related to poor sleep quality compared to abnormal sleep duration. Our findings suggest sleep quality to be a more important indicator for psychological and overall health compared to sleep duration.

## Introduction

Sleep plays an essential role in the health and well-being throughout one's life. Getting enough good quality sleep is necessary for physiologic restoration and recovery, and the lack of it has

Institute of Mental Health. Data is not available for online access, however, readers who wish to gain access to the data can write to the Clinical Research Committee, Institute of Mental Health/ Woodbridge Hospital Secretariat at IMHRESEARCH@imh.com.sg. Access can be granted subject to the Institutional Review Board (IRB) and the research collaborative agreement guidelines. This is a requirement mandated for this research study by our IRB and funders.

**Funding:** This research is supported by the Ministry of Health, Singapore (https://www.moh. gov.sg/) and Temasek Innovates (https://www. temasekfoundation-innovates.org.sg/). The funders had no role in study design, data collection and analysis, decision to publish, or preparation of the manuscript.

**Competing interests:** The authors have declared that no competing interests exist.

been identified as a growing public health concern. Just as adverse sleep issues can increase the risk of health problems, several diseases and disorders can also affect the amount and quality of sleep in individuals. In perhaps the earliest study that looked at the relationship between sleep and health, Hammond [1] observed those who had 7 hours of sleep to report the lowest mortality during a 2-year follow up, with increased death rates found among those who reported shorter or longer sleep duration [1]. The results from this study appear to have driven subsequent research on sleep duration and physical health. For example, several cohort studies have focused on the relationship between sleep duration with cardiovascular- and cancer- specific outcomes and all-cause mortality [2–5]. Research on sleep quality only began to gain more attention after Ford and Kamerow [6] found insomnia to greatly increase the risk of psychiatric disorders. The conceptualization of sleep as two distinct constructs of duration and quality has since been recognised and despite having some extent of overlap, there are qualitative differences between them. "Sleep duration" refers to the total amount of sleep obtained, either during the nocturnal sleep episode or across the 24-h period [7] while "sleep quality" includes the quantitative aspects of sleep such as sleep quantity, sleep latency, or number of arousals at night, as well as the largely subjective indices of sleep, such as depth of sleep, how well rested one feels upon awakening and general satisfaction with sleep [8].

While many studies have since examined the link between sleep and chronic diseases, Bin [9] identified several limitations in this field of research. Firstly, it has been proposed by Bin that much of these evidence points to connecting sleep duration to physical health, and linking sleep quality to mental health. While several meta-analyses have found significant associations between insomnia and mortality and cardiovascular diseases [10–12], there remains a dearth of research looking at the relationship between sleep disturbance and other physical diseases, or that between sleep duration and psychiatric diseases. Secondly, Bin also highlighted that sleep duration and quality have been conceptualized so distinctly that many have failed to recognise that they are measures of the same underlying phenomenon [9]. As a result, nationally representative data on both sleep quality and sleep duration are rarely available from the same source. While few studies conducted in large population samples may have examined the individual association of the two sleep measures with both physical and mental health [13–17], mental health was mainly evaluated only at symptomatic level such as the use of emotional functioning, perceived stress, and severity of depression and anxiety measured on screening instruments.

As noted previously, sleep characteristics in general populations have been studied; typically by assessing sleep duration and quality as the outcomes [18–21]. Based on recommendations by the National Sleep Foundation, the age-appropriate sleep duration was suggested to be 7 to 9 hours for an healthy younger adult or adult with normal sleep, and 7 to 8 hours of sleep for an older adult [22]. Perceived good sleep quality, on the other hand, is characterized by subjective reports of the continuity and restfulness of sleep, including absence of significant sleep disturbances. The roles of sleep duration and sleep quality are believed to be inextricably linked; people with short and long sleep tend to be those who also report sleep disturbance [23, 24]. In other words, the short and long sleep durations may reflect poor sleep quality beyond absolute sleep hours. Yet, other evidence also suggested that sleep quality may not be synonymous with sleep duration. It has been found that the reported usual sleep durations among groups who complain of insomnia and sleeping pill use were well within the range of those without sleep problems [25], while individuals with poor sleep quality were also found to have sufficient sleep [26]. For this reason, it may be more appropriate for studies that include sleep duration or sleep quality as the variable of interest to adjust for the effect of each other in respective analysis.

To address the above limitations and to elucidate the relationship between sleep and chronic diseases, this study therefore aimed to examine the associations of sleep duration and

sleep quality (independently and combined) with lifetime or 12-month physical and mental health diagnoses, using data from a national, epidemiological survey.

## Materials and methods

### Study overview

Data was collected as part of the Singapore Mental Health Study (SMHS) 2016, the second epidemiological survey conducted to establish the prevalence of specific mental illnesses and their associated factors among adult residents (citizen and permanent residents) aged ≥18 years in Singapore [27]. Approval of study was obtained from the institutional review board- National Healthcare Group, Domain Specific Review Board, Singapore. Field interviewers were required to undergo a two-week structured training program conducted by the research team members from the Institute of Mental Health, Singapore, who had been trained and certified by the official World Mental Health Composite International Diagnostic Interview (WMH-CIDI) Training and Research Centre at the University of Michigan. Those who did not meet the standard of knowledge and competency were not allowed to proceed with the fieldwork, which was held between the period of August 2016 to April 2018. Computer-assisted personal interviews (CAPIs) were conducted face-to-face by the interviewers in any of the three preferred languages: English, Chinese and Malay.

### Sample

The survey was designed to be representative of adult citizens including Singaporeans and Permanent Residents aged 18 years and above. The respondents were randomly selected from a national population registry that maintains the names and sociodemographic information of all Singapore residents with 16 strata defined accordingly to ethnicity (Chinese, Malay, Indian, Others) and age groups (18–34, 35–49, 50–64, 65 and above). To ensure an adequate sample in the minority groups and improve the precision for subgroup analysis, a disproportionate stratified sampling (by age and ethnicity) was adopted; where those aged 65 years and above, Malays and Indians were oversampled. Those residents who were incapable of doing an interview due to severe physical or mental conditions, language barriers, living outside the country, institutionalized or hospitalized at the time of the survey, and those were not contactable due to incomplete or incorrect address, were determined as ineligibles and were excluded from the survey. All study participants and the legal representatives for those aged below 21 years of age provided written informed consent prior to the study. The detailed methodology of the SMHS has been described elsewhere [28].

### Data collection

Information such as age, gender, ethnicity, marital status, education, employment status, household income was collected using a structured questionnaire. The height and weight of each respondent were measured to calculate their body mass index (BMI) for risk assessment of cardiovascular disease. Participants were also asked of their current smoking status (smoker, ex-smoker or non-smoker) and how often did they usually have any kind of alcoholic drink during the past 12 months. These sociodemographic and lifestyle data were treated as the main covariates.

### Sleep measures

Although sleep quality is a widely accepted clinical construct, it represents a complex phenomenon that is not readily defined and difficult to measure objectively. To better quantify quality

of sleep, the Pittsburg Sleep Quality Index (PSQI), a reliable and validated standardized measure has been developed to assess sleep quality and disturbance over the 'past month' [29]. This 19-item self-reported questionnaire generates seven component scores: subjective sleep quality, latency, duration, habitual sleep efficiency, sleep disturbances, use of sleep medications, and day-time dysfunction, as well as a global score that ranges from 0 to 21 and discriminates between "good" and "poor" sleepers. Although sleep duration (actual hours spent sleeping) is a component of the sleep quality, they are universally recognised as two distinct constructs. The relationships between sleep quality, and measures of health, well-being and sleepiness have been found to be independent of any effect by sleep quantity [8].

In the current study, we attempted to look at the independent and combined associations of sleep quality and sleep duration with various health conditions. For sleep quality, a global PSQI score of ≥5 was considered indicative of a poor sleep quality [29]. For sleep duration, we adopted similar categorization as previous studies [15, 30] and classified actual sleep duration into average ≤6 hours, 7–8 hours and ≥9 hours per day. The category of 7-8h/day was chosen as the reference for sleep duration to capture possible non-linear relationship between sleep duration and its associated variables, and for the purpose of comparison across studies. For the combined sleep variable (i.e., "duration + quality"), 7-8h/day & good sleep was selected as the reference category from the six available combined levels. The Cronbach's alpha for the seven components of the PSQI were 0.604, 0.701, and 0.633, respectively for the English, Chinese and Malay administered versions.

## Physical disorders

Respondents were asked if they ever had any of the listed major health problems using a modified version of the CIDI 3.0 checklist of chronic medical disorders. The list of 18 medical disorders were (1) asthma, (2) high blood sugar or diabetes, (3) hypertension and high blood pressure, (4) arthritis or rheumatism, (5) cancer diagnosis, (6) a neurological condition, such as epilepsy, convulsions, (7) Parkinson's disease, (8) stroke or major paralysis (inability to use arms or walk), (9) congestive heart failure, (10) heart diseases including a heart attack, coronary heart diseases, angina, or other heart disease, (11) back problems including disk or spine, (12) stomach ulcer, (13) chronic inflamed bowel, enteritis, or colitis, (14) thyroid disease, (15) kidney failure, (16) migraine headaches, (17) chronic lung diseases such as chronic bronchitis or emphysema, and lastly (18) hyperlipidaemia or high cholesterol. For the purpose of this study, these 18 disorders were regrouped into 9 major categories as reported in Tables 2 and 4. Any physical disorder was defined as the presence of any of the 18 chronic conditions.

## Mental disorders

The diagnoses of mental disorders were established using the WMH-CIDI version 3.0, a fully structured diagnostic instrument based on Diagnostic and Statistical Manual of Mental Disorders, 4th Edition (DSM-IV) and International Classification of Disease, 10th Revision (ICD-10) Classification of Mental and Behavioural Disorders criteria. To reduce respondent burden, participants were only required to complete respective diagnostic section of the questionnaire if they had answered positively to a specific screening question. For the purpose of this study, only select mental health modules were administered to determine diagnoses of major depressive disorder (MDD), bipolar disorder (BD), dysthymic disorder (DD), general anxiety disorder (GAD), obsessive compulsive disorder (OCD) and alcohol use disorder (AUD). Any mental disorder was defined as the presence of any of the 6 mental conditions. In our study, "12-month diagnosis" is indicative of an episode within the last 12-months while "lifetime diagnosis" includes both 12-month and past episode(s) reported in the interview at that visit.

Organic exclusion and diagnostic hierarchy rules were applied to generate the final diagnoses. We have examined both the lifetime and 12-month prevalence to explore possible difference in the relationship between the sleep variables and mental disorders.

## Statistical analyses

Data was analysed by IBM SPSS Complex Samples, version 23.0 and estimates were weighted to adjust for over-sampling and non-response. Individual weights were also post-stratified by age and ethnicity according to the Singapore residential population statistics in 2014. Descriptive statistics was tabulated for the overall sample. A series of regression models were conducted such that the estimated odds ratios (ORs) measure the strength of association between each disorder and each of the three sleep variables (sleep quality, sleep duration and "quality + duration") while controlling for other confounding variables. In these regression models (one for physical disorders, one for 12-month history of mental disorders and one for lifetime history of mental disorders), the sleep measure of interest was treated as a dependent variable while mental and physical disorders were treated as main independent variables. For example, in assessing the independent association of sleep quality with physical health, sleep quality was regressed on all the physical disorders in a single model while including sociodemographic (age group, gender, ethnicity, marital status, education and household income), lifestyle factors (BMI, smoking and drinking statues), sleep duration and the presence of any mental disorder as covariates. By doing so, we were able to control for the effect of multimorbidity among the individuals. Two types of regression models were utilized. Binary logistic regressions were conducted when sleep quality (poor vs good) was analysed as the binary dependent variable while multinomial logistic regressions were conducted when sleep duration (≤6h vs 7-8h vs ≥9h/day) and "duration + quality" were analysed as the dependent variables with three categories. Statistical significance was set at p<0.05 level using two-sided tests. As this study was exploratory in nature, corrections for multiple comparisons were not performed.

## Results

A total of 6,126 residents were interviewed for the SMHS 2016 study. Table 1 shows the demographic distribution of the total sample, stratified by sleep quality and sleep duration (see S1 Table for the distribution of physical and mental disorders, stratified by sleep quality and sleep duration).

### Associations of each independent sleep variable with physical disorders

Table 2 shows the associations between chronic physical disorders and each independent sleep variable. After adjusting for sociodemographic/ lifestyle factors, all other physical disorders, any lifetime mental disorder and the other sleep measure, those who had poor sleep (vs good sleep; OR = 1.6, 95% CI 1.3–2.0) and those who slept ≤6h/day (vs 7-8h/day; OR = 1.4, 95% CI 1.1–1.7) in the past month were both significantly associated with higher odds of having a chronic pain condition. Further analysis involving the individual pain conditions revealed only migraine headaches (OR = 1.6, 95% CI 1.2–2.2, p = 0.002) to be significantly associated with short sleep while both migraine headaches (OR = 1.8, 95% CI 1.3–2.4, p<0.001) and back problems (OR = 1.4 95% CI 1.1–1.9, p = 0.018) were significantly associated with poor sleep. Poor sleep quality was also associated with having higher odds of having any physical disorder (OR = 1.4, 95% CI 1.1–1.7).

### Associations of each independent sleep variable with mental disorders

Table 3 shows the associations between lifetime and 12-month mental disorders with each independent sleep variable. After adjusting for sociodemographic/ lifestyle factors, all other

**Table 1. Sociodemographic and sleep profile of population (n = 6,126).**

| | | N | Weighted % | Sleep Quality | | | | Sleep Duration | | | | | |
| | | | | Poor | | Good | | ≤6hrs | | 7-8hrs | | ≥9hrs | |
| | | | | n | % | n | % | n | % | n | % | n | % |
|---|---|---|---|---|---|---|---|---|---|---|---|---|---|
| Age Group (years) | 18–34 | 1707 | 30.4 | 704 | 41.5 | 992 | 58.5 | 803 | 47.1 | 790 | 46.3 | 112 | 6.6 |
| | 35–49 | 1496 | 29.6 | 507 | 34.1 | 979 | 65.9 | 757 | 50.7 | 690 | 46.2 | 47 | 3.1 |
| | 50–64 | 1626 | 26.9 | 580 | 36.0 | 1032 | 64.0 | 877 | 54.0 | 695 | 42.8 | 51 | 3.1 |
| | 65+ | 1297 | 13.1 | 528 | 41.2 | 755 | 58.8 | 650 | 50.4 | 556 | 43.1 | 84 | 6.5 |
| Gender | Male | 3068 | 49.6 | 1087 | 35.8 | 1952 | 64.2 | 1547 | 50.6 | 1358 | 44.4 | 155 | 5.1 |
| | Female | 3058 | 50.4 | 1232 | 40.6 | 1806 | 59.4 | 1540 | 50.5 | 1373 | 45.0 | 139 | 4.6 |
| Ethnicity | Chinese | 1782 | 75.7 | 592 | 33.5 | 1176 | 66.5 | 756 | 42.5 | 891 | 50.1 | 132 | 7.4 |
| | Malay | 1990 | 12.5 | 816 | 41.3 | 1161 | 58.7 | 1220 | 61.6 | 693 | 35.0 | 69 | 3.5 |
| | Indian | 1844 | 8.7 | 718 | 39.3 | 1108 | 60.7 | 871 | 47.3 | 887 | 48.2 | 83 | 4.5 |
| | Others | 510 | 3.1 | 193 | 38.1 | 313 | 61.9 | 240 | 47.1 | 260 | 51.0 | 10 | 2.0 |
| Marital status | Married | 3843 | 59.8 | 1348 | 35.4 | 2465 | 64.6 | 1950 | 50.8 | 1733 | 45.2 | 152 | 4.0 |
| | Never married | 1544 | 31.0 | 633 | 41.3 | 898 | 58.7 | 729 | 47.3 | 701 | 45.5 | 112 | 7.3 |
| | Divorced/ separated/widowed | 739 | 9.2 | 338 | 46.1 | 395 | 53.9 | 408 | 55.5 | 297 | 40.4 | 30 | 4.1 |
| Education | Primary & below | 1187 | 16.3 | 481 | 40.8 | 697 | 59.2 | 629 | 53.1 | 479 | 40.5 | 76 | 6.4 |
| | Secondary | 1648 | 23.0 | 646 | 39.5 | 988 | 60.5 | 874 | 53.3 | 685 | 41.8 | 81 | 4.9 |
| | Post-secondary to Pre-university | 1836 | 31.3 | 731 | 40.1 | 1094 | 59.9 | 933 | 50.8 | 802 | 43.7 | 101 | 5.5 |
| | University | 1455 | 29.4 | 461 | 32.0 | 979 | 68.0 | 651 | 44.8 | 765 | 52.7 | 36 | 2.5 |
| Employment | Employed | 4055 | 72.0 | 1454 | 36.1 | 2573 | 63.9 | 2094 | 51.7 | 1828 | 45.1 | 128 | 3.2 |
| | Economically inactive | 1716 | 22.7 | 697 | 41.0 | 1002 | 59.0 | 833 | 48.7 | 748 | 43.7 | 129 | 7.5 |
| | Unemployed | 354 | 5.3 | 168 | 48.0 | 182 | 52.0 | 159 | 45.3 | 155 | 44.2 | 37 | 10.5 |
| Household income (SGD) | Below 2000 | 1147 | 14.8 | 503 | 44.2 | 636 | 55.8 | 590 | 51.7 | 470 | 41.2 | 81 | 7.1 |
| | 2000–3999 | 1331 | 18.0 | 538 | 40.6 | 787 | 59.4 | 735 | 55.3 | 536 | 40.3 | 59 | 4.4 |
| | 4000–5999 | 1113 | 19.2 | 400 | 36.3 | 703 | 63.7 | 596 | 53.6 | 472 | 42.5 | 43 | 3.9 |
| | 6000–9999 | 1003 | 19.6 | 365 | 36.7 | 630 | 63.3 | 478 | 47.7 | 491 | 49.0 | 33 | 3.3 |
| | 10000 and above | 861 | 18.3 | 276 | 32.4 | 575 | 67.6 | 375 | 43.7 | 460 | 53.6 | 24 | 2.8 |
| BMI (kg/m²) | Underweight (<18.5) | 265 | 6.4 | 103 | 39.2 | 160 | 60.8 | 105 | 39.6 | 137 | 51.7 | 23 | 8.7 |
| | Low risk (18.5–22.9) | 1372 | 32.6 | 505 | 36.9 | 862 | 63.1 | 624 | 45.5 | 672 | 49.0 | 76 | 5.5 |
| | Moderate risk (23.0–27.4) | 2000 | 39.7 | 716 | 36.1 | 1267 | 63.9 | 960 | 48.0 | 940 | 47.0 | 98 | 4.9 |
| | High risk (>27.4) | 1642 | 21.3 | 659 | 40.4 | 973 | 59.6 | 955 | 58.2 | 635 | 38.7 | 50 | 3.0 |
| Smoking status | Current smoker | 1180 | 16.0 | 495 | 42.1 | 680 | 57.9 | 642 | 54.5 | 474 | 40.2 | 62 | 5.3 |
| | Ex-smoker | 747 | 10.6 | 307 | 41.3 | 437 | 58.7 | 396 | 53.1 | 304 | 40.8 | 46 | 6.2 |
| | Non-smoker | 4195 | 73.3 | 1516 | 36.5 | 2639 | 63.5 | 2047 | 48.9 | 1952 | 46.6 | 186 | 4.4 |
| Drinking status during the last 12-month | Non-drinker | 3996 | 50.4 | 1511 | 38.0 | 2462 | 62.0 | 2126 | 53.2 | 1679 | 42.0 | 188 | 4.7 |
| | <1 episode per month | 1075 | 28.1 | 395 | 37.0 | 672 | 63.0 | 485 | 45.2 | 534 | 49.7 | 55 | 5.1 |
| | <5 episodes per month | 695 | 14.9 | 269 | 39.0 | 421 | 61.0 | 303 | 43.7 | 360 | 51.9 | 31 | 4.5 |
| | ≥5 episodes per month | 349 | 6.6 | 142 | 41.2 | 203 | 58.8 | 171 | 49.0 | 158 | 45.3 | 20 | 5.7 |

mental disorders, any physical disorder and the other sleep measure, poor (vs good) sleep quality in the past month was significantly associated with having increased odds of having MDD (lifetime: OR = 2.0, 95% CI 1.4–2.9; 12-month: OR = 3.1, 95% CI 1.8–5.4), BD (lifetime: OR = 2.8, 95% CI 1.5–5.2; 12-month: OR = 4.2, 95% CI 1.8–9.4), GAD (lifetime: OR = 2.3, 95% CI 1.2–4.4; 12-month: OR = 4.3, 95% CI 1.8–10.5), and OCD (lifetime: OR = 2.2, 95% CI 1.4–3.4; 12-month: OR = 2.4, 95% CI 1.4–3.9), while having ≤6h/day (vs 7-8h/day) sleep duration in the past month was associated with increased odds of having OCD (lifetime: OR = 1.6, 95% CI 1.0–2.5; 12-month: OR = 2.0, 95% CI 1.2–3.3). In addition, poor sleep quality was

**Table 2. Independent associations between sleep quality/ duration and physical health conditions[1] (n = 5,186).**

| | Sleep Quality[2] | | | | Sleep Duration[3] | | | | | | | |
| | Poor (vs Good) | | | | ≤6hrs (vs 7-8hrs) | | | | ≥9hrs (vs 7-8hrs) | | | |
| | OR | 95% CI | | p-value | OR | 95% CI | | p-value | OR | 95% CI | | p-value |
| | | Lower | Upper | | | Lower | Upper | | | Lower | Upper | |
| Hypertension | 1.086 | 0.843 | 1.401 | 0.523 | 1.125 | 0.880 | 1.439 | 0.347 | 0.987 | 0.546 | 1.787 | 0.966 |
| Hyperlipidaemia | 1.131 | 0.878 | 1.458 | 0.340 | 0.978 | 0.772 | 1.239 | 0.855 | 1.067 | 0.614 | 1.852 | 0.819 |
| Diabetes | 1.173 | 0.874 | 1.574 | 0.288 | 0.977 | 0.722 | 1.321 | 0.879 | 1.031 | 0.575 | 1.850 | 0.918 |
| Asthma | 1.011 | 0.784 | 1.304 | 0.931 | 1.174 | 0.899 | 1.534 | 0.239 | 0.653 | 0.349 | 1.221 | 0.182 |
| Chronic pain[a] | 1.582 | 1.275 | 1.964 | <0.001 | 1.372 | 1.098 | 1.715 | 0.005 | 0.771 | 0.432 | 1.373 | 0.377 |
| Cardiovascular disorders[b] | 0.897 | 0.606 | 1.326 | 0.585 | 0.791 | 0.526 | 1.190 | 0.261 | 0.655 | 0.288 | 1.489 | 0.312 |
| Thyroid diseases | 1.126 | 0.665 | 1.906 | 0.658 | 1.244 | 0.728 | 2.125 | 0.424 | 0.734 | 0.227 | 2.374 | 0.605 |
| Ulcer[c] | 1.701 | 0.901 | 3.211 | 0.101 | 1.164 | 0.597 | 2.269 | 0.655 | 0.879 | 0.185 | 4.168 | 0.871 |
| Cancer | 1.122 | 0.595 | 2.117 | 0.722 | 0.809 | 0.424 | 1.542 | 0.519 | 2.318 | 0.748 | 7.180 | 0.145 |
| Any physical disorder[4] | 1.393 | 1.148 | 1.689 | 0.001 | 1.148 | 0.953 | 1.383 | 0.147 | 0.855 | 0.580 | 1.261 | 0.429 |

[1] Analysis controlled for sociodemographic/ lifestyle factors + all other physical disorders + any lifetime mental disorder

[2] Analyses controlled for sleep duration

[3] Analyses controlled for sleep quality

[4] Analysis controlled for sociodemographic/ lifestyle factors + any mental disorder only (n = 5,242)

[a] Comprises arthritis or rheumatism, back problems including disk or spine problems, migraine headaches

[b] Comprises stroke or major paralysis, heart attack, coronary heart disease, angina, congestive heart failure or other heart disease

[c] Comprises chronic inflamed bowel problems such as stomach ulcer, enteritis or colitis

**Table 3. Independent associations between sleep quality/ duration and mental health conditions[1].**

| | | Sleep Quality[2] | | | | Sleep Duration[3] | | | | | | | |
| | | Poor (vs Good) | | | | ≤6hrs (vs 7-8hrs) | | | | ≥9hrs (vs 7-8hrs) | | | |
| | | OR | 95% CI | | p-value | OR | 95% CI | | p-value | OR | 95% CI | | p-value |
| | | | Lower | Upper | | | Lower | Upper | | | Lower | Upper | |
| Lifetime | MDD | 2.019 | 1.407 | 2.897 | <0.001 | 1.091 | 0.754 | 1.578 | 0.644 | 1.252 | 0.593 | 2.643 | 0.555 |
| | DD | 3.572 | 0.686 | 18.606 | 0.131 | 0.542 | 0.169 | 1.742 | 0.304 | 3.055 | 0.282 | 33.142 | 0.358 |
| | BD | 2.765 | 1.483 | 5.155 | 0.001 | 1.226 | 0.627 | 2.398 | 0.552 | 1.068 | 0.251 | 4.539 | 0.929 |
| | GAD | 2.310 | 1.219 | 4.376 | 0.010 | 0.983 | 0.544 | 1.777 | 0.956 | 1.059 | 0.169 | 6.620 | 0.951 |
| | OCD | 2.193 | 1.398 | 3.439 | 0.001 | 1.598 | 1.009 | 2.530 | 0.046 | 0.754 | 0.203 | 2.801 | 0.673 |
| | AUD | 1.009 | 0.646 | 1.577 | 0.967 | 0.932 | 0.583 | 1.489 | 0.768 | 1.158 | 0.415 | 3.232 | 0.780 |
| | Any mental disorder[4] | 2.170 | 1.699 | 2.772 | <0.001 | 1.292 | 1.002 | 1.666 | 0.048 | 0.987 | 0.525 | 1.855 | 0.968 |
| 12-month | MDD | 3.128 | 1.802 | 5.430 | <0.001 | 1.302 | 0.735 | 2.308 | 0.366 | 0.687 | 0.158 | 2.979 | 0.615 |
| | DD | 2.292 | 0.529 | 9.931 | 0.267 | 0.445 | 0.131 | 1.514 | 0.195 | - | - | - | - |
| | BD | 4.154 | 1.842 | 9.367 | 0.001 | 1.529 | 0.700 | 3.341 | 0.286 | 0.107 | 0.010 | 1.170 | 0.067 |
| | GAD | 4.312 | 1.767 | 10.523 | 0.001 | 1.283 | 0.533 | 3.091 | 0.578 | 3.170 | 0.378 | 26.599 | 0.288 |
| | OCD | 2.365 | 1.444 | 3.872 | 0.001 | 1.992 | 1.184 | 3.349 | 0.009 | 1.329 | 0.330 | 5.359 | 0.689 |
| | AUD | 1.320 | 0.579 | 3.013 | 0.509 | 0.474 | 0.172 | 1.307 | 0.149 | 0.801 | 0.124 | 5.173 | 0.816 |
| | Any mental disorder[4] | 3.152 | 2.290 | 4.339 | <0.001 | 1.418 | 1.013 | 1.985 | 0.042 | 0.908 | 0.366 | 2.253 | 0.835 |

[1] Analysis controlled for sociodemographic/ lifestyle factors + all other mental disorders + any physical disorder

[2] Analyses controlled for sleep duration

[3] Analyses controlled for sleep quality

[4] Analyses controlled for sociodemographic/ lifestyle factors + any physical disorder only (n = 5,242)

MDD: major depressive disorder; DD: dysthymic disorder; BD: bipolar disorder; GAD: generalized anxiety disorder; OCD: obsessive compulsive disorder; AUD: alcohol use disorder

associated with having any mental disorder (lifetime: OR = 2.2, 95% CI 1.7–2.8; 12-month: OR = 3.2, 95% CI 2.3–4.3).

## Associations of the combined sleep variable with physical disorders

Table 4 shows the associations between physical disorders with the combined sleep variable. '<6h/day and poor sleep' (vs '7-8h/day and good sleep') was associated with increased odds of having a chronic pain condition (OR = 2.2, 95% CI 1.7–2.9) and any physical disorder (OR = 1.8, 95% CI 1.4–2.2). Further analysis involving the individual pain conditions revealed only migraine headaches (OR = 2.5, 95% CI 1.7–3.7, p<0.001) and back problems (OR = 1.7 95% CI 1.2–2.5, p = 0.006) to be significantly associated with '<6h/day and poor sleep'. Among those having 7-8h/day of sleep, poor (vs good) sleep quality was associated with increased odds of having ulcers and inflamed bowel disorder (OR = 3.1, 95% CI 1.2–8.0).

**Table 4. Combined associations of sleep quality + duration with physical health conditions[1] (n = 5,186).**

| Ref group: 7–8 hrs & good sleep | | OR | 95% CI | | p-value |
|---|---|---|---|---|---|
| | | | Lower | Upper | |
| Hypertension | ≤6h & good sleep | 1.074 | 0.792 | 1.458 | 0.645 |
| | ≥9h & good sleep | 1.128 | 0.565 | 2.249 | 0.733 |
| | ≤6h & poor sleep | 1.267 | 0.915 | 1.755 | 0.154 |
| | 7-8h & poor sleep | 1.028 | 0.680 | 1.556 | 0.895 |
| | ≥9h & poor sleep | 0.736 | 0.266 | 2.041 | 0.556 |
| Hyperlipidaemia | ≤6h & good sleep | 0.906 | 0.679 | 1.208 | 0.500 |
| | ≥9h & good sleep | 1.192 | 0.649 | 2.191 | 0.571 |
| | ≤6h & poor sleep | 1.152 | 0.840 | 1.580 | 0.379 |
| | 7-8h & poor sleep | 1.042 | 0.684 | 1.586 | 0.848 |
| | ≥9h & poor sleep | 0.834 | 0.313 | 2.219 | 0.716 |
| Diabetes | ≤6h & good sleep | 0.740 | 0.492 | 1.112 | 0.147 |
| | ≥9h & good sleep | 1.005 | 0.501 | 2.017 | 0.988 |
| | ≤6h & poor sleep | 1.191 | 0.813 | 1.744 | 0.370 |
| | 7-8h & poor sleep | 0.761 | 0.458 | 1.263 | 0.291 |
| | ≥9h & poor sleep | 0.982 | 0.361 | 2.673 | 0.972 |
| Asthma | ≤6h & good sleep | 1.191 | 0.856 | 1.658 | 0.299 |
| | ≥9h & good sleep | 0.730 | 0.361 | 1.475 | 0.380 |
| | ≤6h & poor sleep | 1.177 | 0.843 | 1.643 | 0.337 |
| | 7-8h & poor sleep | 1.050 | 0.693 | 1.591 | 0.817 |
| | ≥9h & poor sleep | - | - | - | - |
| Chronic pain | ≤6h & good sleep | 1.118 | 0.831 | 1.506 | 0.461 |
| | ≥9h & good sleep | 0.839 | 0.432 | 1.631 | 0.605 |
| | ≤6h & poor sleep | 2.199 | 1.677 | 2.884 | <0.001 |
| | 7-8h & poor sleep | 1.188 | 0.816 | 1.728 | 0.369 |
| | ≥9h & poor sleep | 0.737 | 0.268 | 2.029 | 0.555 |
| Cardiovascular disorder | ≤6h & good sleep | 0.668 | 0.385 | 1.161 | 0.152 |
| | ≥9h & good sleep | 0.341 | 0.117 | 0.992 | 0.048 |
| | ≤6h & poor sleep | 0.703 | 0.426 | 1.161 | 0.168 |
| | 7-8h & poor sleep | 0.543 | 0.264 | 1.116 | 0.097 |
| | ≥9h & poor sleep | 1.514 | 0.416 | 5.507 | 0.529 |

*(Continued)*

**Table 4.** (Continued)

| Ref group: 7–8 hrs & good sleep | | OR | 95% CI | | p-value |
|---|---|---|---|---|---|
| | | | Lower | Upper | |
| Thyroid diseases | ≤6h & good sleep | 1.534 | 0.799 | 2.945 | 0.198 |
| | ≥9h & good sleep | - | - | - | - |
| | ≤6h & poor sleep | 1.371 | 0.732 | 2.568 | 0.324 |
| | 7-8h & poor sleep | 1.657 | 0.740 | 3.711 | 0.219 |
| | ≥9h & poor sleep | - | - | - | - |
| Ulcer | ≤6h & good sleep | 1.844 | 0.800 | 4.248 | 0.151 |
| | ≥9h & good sleep | - | - | - | - |
| | ≤6h & poor sleep | 2.107 | 0.914 | 4.857 | 0.080 |
| | 7-8h & poor sleep | 3.073 | 1.187 | 7.958 | 0.021 |
| | ≥9h & poor sleep | - | - | - | - |
| Cancer | ≤6h & good sleep | 0.578 | 0.227 | 1.474 | 0.251 |
| | ≥9h & good sleep | 3.281 | 0.993 | 10.840 | 0.051 |
| | ≤6h & poor sleep | 1.127 | 0.527 | 2.413 | 0.757 |
| | 7-8h & poor sleep | 0.992 | 0.323 | 3.045 | 0.988 |
| | ≥9h & poor sleep | - | - | - | - |
| Any physical disorder[2] | ≤6h & good sleep | 0.919 | 0.733 | 1.151 | 0.462 |
| | ≥9h & good sleep | 0.922 | 0.604 | 1.406 | 0.705 |
| | ≤6h & poor sleep | 1.761 | 1.398 | 2.218 | <0.001 |
| | 7-8h & poor sleep | 0.999 | 0.735 | 1.357 | 0.993 |
| | ≥9h & poor sleep | 0.686 | 0.292 | 1.612 | 0.387 |

[1] Analysis controlled for sociodemographic/ lifestyle factors + all other physical disorders + any lifetime mental disorder; data not presented for all due to low cell sizes;
Ref group: 7–8 hrs & good sleep

[2] Analysis controlled for sociodemographic/ lifestyle factors + any mental disorder only (n = 5,242)

## Associations of the combined sleep variable with mental disorders

Table 5 shows the associations between lifetime mental disorders with the combined sleep variable. Compared to '7-8h/day and good sleep', all three sleep duration groups (≤6h & 7-8h & ≥9h/day) with poor sleep in the past month were associated with higher chance of having lifetime MDD and any mental disorder. '≤6h/day and poor sleep' was also associated with having lifetime BD (OR = 3.8, 95% CI 1.3–11.6) and OCD (OR = 3.7, 95% CI 2.0–7.1).

Table 6 shows the associations between 12-month mental disorders with the combined sleep variable. '≤6h/day and poor sleep' (vs '7-8h/day and good sleep') in the past month was associated with higher chance of having lifetime MDD (OR = 3.5, 95% CI 1.7–7.1), BD (OR = 4.4, 95% CI 1.4–13.6), GAD (OR = 3.8, 95% CI 1.2–11.9), OCD (OR = 4.3, 95% CI 2.3–8.3), and any mental disorder (OR = 3.9, 95% CI 2.6–5.9).

## Discussion

In the current study, we assessed the associations of both sleep duration and sleep quality (independently and combined) with comorbid physical and mental disorders established using CIDI among a large pool of community-dwelling adults. By using a population cohort of healthy adults and investigating the relationship between sleep and diseases across multiple health domains in the same sample, we can circumvent challenges associated with studying clinical populations and provide new insights.

**Table 5. Combined associations of sleep quality + duration with lifetime mental health conditions[1] (n = 5,242).**

| Ref group: 7–8 hrs & good sleep | | OR | 95% CI | | p-value |
|---|---|---|---|---|---|
| | | | Lower | Upper | |
| MDD | ≤6h & good sleep | 1.095 | 0.627 | 1.913 | 0.749 |
| | ≥9h & good sleep | 1.157 | 0.463 | 2.888 | 0.755 |
| | ≤6h & poor sleep | 2.204 | 1.343 | 3.617 | 0.002 |
| | 7-8h & poor sleep | 2.094 | 1.199 | 3.656 | 0.009 |
| | ≥9h & poor sleep | 3.025 | 1.626 | 5.630 | <0.001 |
| BD | ≤6h & good sleep | 0.623 | 0.223 | 1.741 | 0.367 |
| | ≥9h & good sleep | - | - | - | - |
| | ≤6h & poor sleep | 3.821 | 1.254 | 11.643 | 0.018 |
| | 7-8h & poor sleep | 1.171 | 0.236 | 5.799 | 0.847 |
| | ≥9h & poor sleep | - | - | - | - |
| GAD | ≤6h & good sleep | 0.622 | 0.190 | 2.038 | 0.433 |
| | ≥9h & good sleep | - | - | - | - |
| | ≤6h & poor sleep | 2.203 | 0.823 | 5.898 | 0.116 |
| | 7-8h & poor sleep | 1.007 | 0.286 | 3.547 | 0.991 |
| | ≥9h & poor sleep | - | - | - | - |
| OCD | ≤6h & good sleep | 1.179 | 0.527 | 2.637 | 0.688 |
| | ≥9h & good sleep | - | - | - | - |
| | ≤6h & poor sleep | 3.739 | 1.981 | 7.059 | <0.001 |
| | 7-8h & poor sleep | 1.466 | 0.625 | 3.437 | 0.379 |
| | ≥9h & poor sleep | - | - | - | - |
| AUD | ≤6h & good sleep | 1.261 | 0.698 | 2.279 | 0.441 |
| | ≥9h & good sleep | 0.956 | 0.372 | 2.457 | 0.926 |
| | ≤6h & poor sleep | 0.837 | 0.482 | 1.452 | 0.526 |
| | 7-8h & poor sleep | 1.539 | 0.826 | 2.868 | 0.175 |
| | ≥9h & poor sleep | 2.435 | 0.864 | 6.867 | 0.092 |
| Any mental disorder[2] | ≤6h & good sleep | 1.044 | 0.734 | 1.486 | 0.810 |
| | ≥9h & good sleep | 0.797 | 0.367 | 1.732 | 0.567 |
| | ≤6h & poor sleep | 2.741 | 2.012 | 3.733 | <0.001 |
| | 7-8h & poor sleep | 1.622 | 1.099 | 2.393 | 0.015 |
| | ≥9h & poor sleep | 2.749 | 1.070 | 7.063 | 0.036 |

[1] Analyses controlled for sociodemographic/ lifestyle factors + all other lifetime mental disorders + any physical disorder; data not presented for all (including dysthymia) due to low cell sizes; 7–8 hrs & good sleep assigned as reference group.

[2] Analysis controlled for sociodemographic/ lifestyle factors + any physical disorder only (n = 5,242)

MDD: major depressive disorder; BD: bipolar disorder; GAD: generalized anxiety disorder; OCD: obsessive compulsive disorder; AUD: alcohol use disorder

By examining sleep duration alongside with sleep quality and their relationships with health statuses from the same population, the current study may be able to provide some form of comparative evidence to determine which of these two sleep indicators may be more important for public health, based on independent associations. The associated physical comorbidities were found to be similar for poor sleep quality and short sleep duration, with only slightly higher odds ratios reported for the former. Sleep quality, but not sleep duration, was found to be associated with any physical disorder(s). However, higher number of mental comorbidities (both lifetime and 12-month) exhibited significant associations with poor sleep quality with larger effect sizes compared to those with short sleep duration. Our findings may imply that sleep disturbance may be associated more strongly with (1) psychological health, and (2) overall health compared to sleep quantity.

**Table 6. Combined associations of sleep quality + duration with 12-month mental health conditions[1] (n = 5,242).**

| Ref group: 7–8 hrs & good sleep | | OR | 95% CI | | p-value |
|---|---|---|---|---|---|
| | | | Lower | Upper | |
| MDD | ≤6h & good sleep | 0.742 | 0.275 | 2.002 | 0.556 |
| | ≥9h & good sleep | - | - | - | - |
| | ≤6h & poor sleep | 3.512 | 1.746 | 7.062 | <0.001 |
| | 7-8h & poor sleep | 2.196 | 0.897 | 5.380 | 0.085 |
| | ≥9h & poor sleep | - | - | - | - |
| BD | ≤6h & good sleep | 0.484 | 0.124 | 1.889 | 0.296 |
| | ≥9h & good sleep | - | - | - | - |
| | ≤6h & poor sleep | 4.402 | 1.421 | 13.632 | 0.010 |
| | 7-8h & poor sleep | 2.877 | 0.809 | 10.229 | 0.103 |
| | ≥9h & poor sleep | - | - | - | - |
| GAD | ≤6h & good sleep | 0.307 | 0.079 | 1.197 | 0.089 |
| | ≥9h & good sleep | - | - | - | - |
| | ≤6h & poor sleep | 3.758 | 1.192 | 11.855 | 0.024 |
| | 7-8h & poor sleep | 1.333 | 0.291 | 6.104 | 0.712 |
| | ≥9h & poor sleep | - | - | - | - |
| OCD | ≤6h & good sleep | 1.464 | 0.638 | 3.362 | 0.368 |
| | ≥9h & good sleep | - | - | - | - |
| | ≤6h & poor sleep | 4.312 | 2.250 | 8.264 | <0.001 |
| | 7-8h & poor sleep | 1.660 | 0.693 | 3.975 | 0.255 |
| | ≥9h & poor sleep | - | - | - | - |
| AUD | ≤6h & good sleep | 0.429 | 0.098 | 1.883 | 0.262 |
| | ≥9h & good sleep | - | - | - | - |
| | ≤6h & poor sleep | 0.663 | 0.247 | 1.782 | 0.415 |
| | 7-8h & poor sleep | 1.477 | 0.446 | 4.890 | 0.523 |
| | ≥9h & poor sleep | - | - | - | - |
| Any mental disorder[2] | ≤6h & good sleep | 0.724 | 0.411 | 1.277 | 0.265 |
| | ≥9h & good sleep | 0.649 | 0.192 | 2.188 | 0.485 |
| | ≤6h & poor sleep | 3.918 | 2.603 | 5.899 | <0.001 |
| | 7-8h & poor sleep | 1.696 | 0.976 | 2.947 | 0.061 |
| | ≥9h & poor sleep | 2.797 | 0.831 | 9.407 | 0.097 |

[1] Analyses controlled for sociodemographic/ lifestyle factors + all other 12-month mental disorders + any physical disorder; data not presented for all (including dysthymia) due to low cell sizes; 7–8 hrs & good sleep assigned as reference group.

[2] Analysis controlled for sociodemographic/ lifestyle factors + any physical disorder only (n = 5,242)

MDD: major depressive disorder; BD: bipolar disorder; GAD: generalized anxiety disorder; OCD: obsessive compulsive disorder; AUD: alcohol use disorder

Other studies have reported similar findings with regard to the importance of sleep quality over sleep duration for mental health. Gadie and colleagues [16] found the strongest association between sleep quality and mental health but only moderate relationships with physical and cognitive health, and such relationships observed were mostly stable across the adult life-span. Among older adults, chronic insomnia symptoms were also associated with worse mental (difference -6.9; SE = 0.4) and physical (difference -2.8; SE = 0.4) well-being, while both recurrent long and short sleep were only associated with physical (difference -3.5; SE = 0.9) well-being [17]. Lastly, poor sleep quality but not sleep duration was found to have significant impact on the development of both depressive symptoms and suicidal ideation among Japanese university freshmen [31]. Despite the fact that research has consistently found short and long sleep to be associated with adverse health outcomes [32, 33], there seems to be a lack of

studies that have provided support for the stronger role of sleep duration over sleep quality in its relation with physical health as Bin proposed [9]. In fact, in one study that examined how onset of impaired sleep affects the risk of established cardiovascular diseases, Clark et al. [34] found the onset of sleep disturbances rather than short or long sleep to predict subsequent risk of hypertension and dyslipidaemia. Although our study did not establish such association(s) between sleep quality and the specific physical conditions, the study by Clarke and colleagues does demonstrate that the quality of sleep may appear to be a more important risk factor for predicting overall health and diseases compared to the quantity of sleep. In studies conducted among college students, average sleep quality was also found to be better related to health, affect balance, satisfaction with life, and feelings of tension, depression, anger, fatigue, sleepiness and confusion than average sleep quantity [8, 35].

In terms of the associations of the combined sleep variable with health statues, our findings revealed poor sleep quality and short (<6h/day) sleep to be associated with the largest number of physical and mental disorders among the remaining five combinations when compared to good sleep quality and mid-range (7-8h/day) sleep duration. The majority of the studies that examined similar interaction effects or associations of the combined sleep variable have consistently found poor sleep quality combined with short duration to have the greatest association with hypertension [36], diabetes or impaired fasting glucose [37–39], psychological distress [40], coronary heart diseases risk and cardiovascular disease mortality [41–43] among various populations. We also found poor sleep quality to be consistently associated with lifetime mental disorders, particularly MDD, regardless of the sleep duration. Poor sleep quality in combination with short, mid-range and long sleep was also found to have robust associations with worse physical, emotional and social functioning in a general population [13]. This further supports that sleep quality may more strongly associated with health conditions than sleep duration.

We did not find any association between any of the sleep measures with physical conditions such as metabolic and cardiovascular diseases despite significant associations exhibited in the vast literature [32–34, 36, 38, 39, 41–43]. The lack of such findings in our study may be due to the confounding effects from other physical conditions and of which, only chronic pain (including arthritis or rheumatism, back problems including disk or spine problems, migraine headaches) was found to be significantly associated with both the sleep variables. Only few population-based studies have included such an extensive range of chronic diseases [15, 30]. In the study of the Korean population, only osteoarthritis and cancer (but not any metabolic or cardiovascular disorders) were found to be associated with short (<6h/day) sleep duration (compared to 7-8h/day) [15]. In the Brazil population, heart disease and vascular problems (but not metabolic conditions such as hypertension and diabetes) were found to be associated with sleep duration [30]. Additionally, other pain disorders such as rheumatism/arthritis/ arthrosis, osteoporosis and back pain/problems were similarly found to be associated with short sleep compared to mid-range sleep duration [30]. In terms of mental disorders in the independent associations, we found poor sleep quality to be associated with almost all mood and anxiety disorders except dysthymia. Associations between depressive and anxiety symptoms or disorders with poor sleep quality were also well-documented in other studies [6, 15, 16, 30, 44, 45]. Furthermore, we observed that these mental disorders continued to exhibit significant associations with sleep quality regardless of whether they were lifetime or 12-month disorders. We also found that except for GAD which becomes significant in terms of its relationship with the combined sleep variable when criterion was changed from lifetime to 12-month diagnoses, the other disorders remained relatively stable in their relationships with both the independent and combined sleep variables across lifetime and 12-month disorders. This further justifies the close relationship between these affective disorders and sleep.

Research has found a possible U-shaped association between sleep duration and health, where short and long sleep (compared to normal or mid-range sleep) were related to an increased risk of morbidity and mortality [16, 30, 33, 46, 47]. The current study, however, did not find any curvilinear trend in all our analyses as the category of >9h/day sleep did not exhibit significant association with any disorder. Furthermore, majority of the odds ratios reported in the associations of the physical disorders with the independent sleep variable (s) were less than 1.0, suggesting that these conditions may be negatively associated with long sleep despite being insignificant. It has been postulated that poor sleep quality may be at least partly responsible for the adverse health outcomes associated with the extremes of sleep duration [9]. Our analyses for the health correlates of sleep duration was controlled for sleep quality and may therefore, explain the absence of significant U-shaped associations. Another possible reason could be the relatively smaller sample size among those who had >9h/day of long sleep (weighted prevalence rate of only 6.2%), thus resulting in lower statistical power to detect any true effect. This has also led to several unreported effect sizes and estimates in the multinomial logistic regression involving associations of the various disorders with the combined sleep variable due to low cell sizes. These findings in our study pertaining to long sleep may therefore need to be interpreted with caution.

The current study used a large nationally representative, community sample of adults from a multi-ethnic population and had addressed several shortcomings of previous studies examining morbidity associated with sleep measures. However, there were still some limitations to this present study. Firstly, although we used the CIDI to establish mental diagnoses, it was not possible to include all mental disorders due to constraints of time, costs, and respondent burden. Secondly, both sleep quality and sleep duration were self-reported; such subjective opinions of participants tend to provide an estimate rather than reflect the actual sleep conditions due to recall bias. Studies have also found that self-reported sleep duration tend to overestimate objectively measured sleep [48, 49] and the extent of over-reporting increased as sleep duration decreased [50]. As a result, short sleepers were at a higher risk of being misclassified as normal or long sleeper, which have possibly led to an inability to detect an increased health risk among such individuals. However, misclassification can lead to bias in either direction (i.e., towards or away from the null). Lastly, causation relationships between sleep and chronic diseases cannot be inferred due to the cross-sectional design of this study. Furthermore, we have used sleep measures in the last one month as a proxy to examine their relationship with lifetime and 12-month health conditions and hence, only associations between sleep and health could be established, and we would have to rule out any impact of sleep on health outcomes which would be of greater concern in the medical research. There is a need to conduct longitudinal studies to shed more light on the directions of influence between these variables.

## Conclusions

This study examined the associations of two sleep variables–sleep duration and sleep quality (independently and combined) with chronic morbidities. Both physical and mental health domains were examined and diagnoses were collected using the validated CIDI. Our findings indicated that sleep quality may be a more important indicator for psychological and overall health compared to sleep duration. We did not observe a U-shaped relationship between sleep duration and morbidity, as only short sleep was associated with adverse health conditions. Poor sleep quality combined with short sleep was associated with the highest number of morbidities and hence, there is a need for public awareness on the relationship between sleep and health.

## Supporting information

**S1 Table. Distribution of physical and mental disorders, stratified by sleep duration and sleep quality.** MDD: major depressive disorder; DD: dysthymic disorder; BD: bipolar disorder; GAD: generalized anxiety disorder; OCD: obsessive compulsive disorder; AUD: alcohol use disorder.
(DOCX)

## Author Contributions

**Conceptualization:** Siow Ann Chong, Janhavi Ajit Vaingankar, Mythily Subramaniam.

**Formal analysis:** Xiao Wei Tan.

**Funding acquisition:** Mythily Subramaniam.

**Methodology:** Siow Ann Chong, Edimansyah Abdin, Mythily Subramaniam.

**Project administration:** Janhavi Ajit Vaingankar, Saleha Shafie, Boon Yiang Chua.

**Software:** Boon Yiang Chua.

**Supervision:** Mythily Subramaniam.

**Writing – original draft:** Lee Seng Esmond Seow.

**Writing – review & editing:** Siow Ann Chong, Janhavi Ajit Vaingankar, Edimansyah Abdin, Saleha Shafie, Boon Yiang Chua, Derrick Heng, Mythily Subramaniam.

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
