## [Decision Letter · Decision Letter 0]

27 May 2020

PONE-D-19-31356

Independent and combined associations of sleep duration and sleep quality with lifetime or 12-month experience of common physical and mental disorders: Results from a multi-ethnic population-based cross-sectional survey

PLOS ONE

Dear Dr. Seow,

Thank you for submitting your manuscript to PLOS ONE. After careful consideration, we feel that it has merit but does not fully meet PLOS ONE’s publication criteria as it currently stands. Therefore, we invite you to submit a revised version of the manuscript that addresses the points raised during the review process.

I am returning your manuscript with three reviews. The reviewers came to different conclusions

about the paper, as you will see. After reading the reviews and looking at your manuscript, I have to concur that your manuscript requires a major revision. Please pay great attention to the following reviewers suggestions and give them due consideration, especially about the uncited meta-analyses on different aspects of the realation between sleep duration/quality and health provided by a reviewer, and several issues through various sections of the manuscript.

We look forward to receiving your revised manuscript.

Kind regards,

Claudio Andaloro

Academic Editor

PLOS ONE

Journal Requirements:

Reviewers' comments:

Reviewer's Responses to Questions

**Comments to the Author**

1. Is the manuscript technically sound, and do the data support the conclusions?

Reviewer #1: Partly

Reviewer #2: Yes

Reviewer #3: Yes

2. Has the statistical analysis been performed appropriately and rigorously? 

Reviewer #1: N/A

Reviewer #2: Yes

Reviewer #3: Yes

3. Have the authors made all data underlying the findings in their manuscript fully available?

Reviewer #1: No

Reviewer #2: No

Reviewer #3: Yes

4. Is the manuscript presented in an intelligible fashion and written in standard English?

Reviewer #1: Yes

Reviewer #2: Yes

Reviewer #3: Yes

5. Review Comments to the Author

Reviewer #1: The background is largely and purposely limited.

These are uncited meta-analyses (sic) that -in different way- assess the relation between sleep duration and quality and health otcomes.

da Silva AA, de Mello RG, Schaan CW, Fuchs FD, Redline S, Fuchs SC. Sleep duration and mortality in the elderly: a systematic review with meta-analysis. BMJ Open. 2016 Feb 17;6(2):e008119. doi: 10.1136/bmjopen-2015-008119.

Gallicchio L, Kalesan B. Sleep duration and mortality: a systematic review and meta-analysis. J Sleep Res. 2009 Jun;18(2):148-58. doi: 10.1111/j.1365-2869.2008.00732.x.

Shen X, Wu Y, Zhang D. Nighttime sleep duration, 24-hour sleep duration and risk of all-cause mortality among adults: a meta-analysis of prospective cohort studies. Sci Rep. 2016 Feb 22;6:21480. doi: 10.1038/srep21480.

Kawada T1. Total sleep time and all cancer mortality: a meta-analysis. Sleep Med. 2020 Apr;68:96. doi: 10.1016/j.sleep.2019.12.029. Epub 2020 Jan 10.

Ge L1, Guyatt G2, Tian J3, Pan B4, Chang Y2, Chen Y4, Li H4, Zhang J4, Li Y4, Ling J3, Yang K5.

Insomnia and risk of mortality from all-cause, cardiovascular disease, and cancer: Systematic review and meta-analysis of prospective cohort studies. Sleep Med Rev. 2019 Dec;48:101215. doi: 10.1016/j.smrv.2019.101215.

Kwok CS1, Kontopantelis E2, Kuligowski G3, Gray M3, Muhyaldeen A4, Gale CP5, Peat GM6, Cleator J7, Chew-Graham C6, Loke YK8, Mamas MA1. Self-Reported Sleep Duration and Quality and Cardiovascular Disease and Mortality: A Dose-Response Meta-Analysis. J Am Heart Assoc. 2018 Aug 7;7(15):e008552. doi: 10.1161/JAHA.118.008552.

Yin J et al. J Relationship of Sleep Duration With All-Cause Mortality and Cardiovascular Events: A Systematic Review and Dose-Response Meta-Analysis of Prospective Cohort Studies. Am Heart Assoc. (2017)

Jike M et al. Long sleep duration and health outcomes: A systematic review, meta-analysis and meta-regression. Sleep Med Rev. (2018)

Li Y, Cai S, Ling Y, Mi S, Fan C, Zhong Y, Shen Q. Association between total sleep time and all cancer mortality: non-linear dose-response meta-analysis of cohort studies.

Ma QQ, Yao Q, Lin L, Chen GC, Yu JB. Sleep duration and total cancer mortality: a meta-analysis of prospective studies. Sleep Med. 2016 Nov - Dec;27-28:39-44. doi: 10.1016/j.sleep.2016.06.036.

Stone CR, Haig TR, Fiest KM, McNeil J, Brenner DR, Friedenreich CM. The association between sleep duration and cancer-specific mortality: a systematic review and meta-analysis. Sleep Med. 2019 Aug;60:211-218. doi: 10.1016/j.sleep.2019.03.026.

Lovato N, Lack L. Insomnia and mortality: A meta-analysis. Cancer Causes Control. 2019 May;30(5):501-525. doi: 10.1007/s10552-019-01156-4. Epub 2019 Mar 22.

García-Perdomo HA, Zapata-Copete J, Rojas-Cerón CA. Sleep duration and risk of all-cause mortality: a systematic review and meta-analysis. Sleep Med Rev. 2019 Feb;43:71-83. doi: 10.1016/j.smrv.2018.10.004.

Krittanawong C, Tunhasiriwet A, Wang Z, Zhang H, Farrell AM, Chirapongsathorn S, Sun T, Kitai T, Argulian E. Association between short and long sleep durations and cardiovascular outcomes: a systematic review and meta-analysis. Epidemiol Psychiatr Sci. 2019 Oct;28(5):578-588. doi: 10.1017/S2045796018000379.

Jike M, Itani O, Watanabe N, Buysse DJ, Kaneita Y. Long sleep duration and health outcomes: A systematic review, meta-analysis and meta-regression. Eur Heart J Acute Cardiovasc Care. 2019 Dec;8(8):762-770. doi: 10.1177/2048872617741733.

Itani O, Jike M, Watanabe N, Kaneita Y. Short sleep duration and health outcomes: a systematic review, meta-analysis, and meta-regression. Sleep Med Rev. 2018 Jun;39:25-36. doi: 10.1016/j.smrv.2017.06.011.

Li W, Wang D, Cao S, Yin X, Gong Y, Gan Y, Zhou Y, Lu Z. Sleep duration and risk of stroke events and stroke mortality: A systematic review and meta-analysis of prospective cohort studies. Int J Cardiol. 2016 Nov 15;223:870-876. doi: 10.1016/j.ijcard.2016.08.302.

Liu TZ, Xu C, Rota M, Cai H, Zhang C, Shi MJ, Yuan RX, Weng H, Meng XY, Kwong JS, Sun X. Sleep duration and risk of all-cause mortality: A flexible, non-linear, meta-regression of 40 prospective cohort studies. Sleep Med Rev. 2017 Apr;32:28-36. doi: 10.1016/j.smrv.2016.02.005.

Yang X, Chen H, Li S, Pan L, Jia C. Association of Sleep Duration with the Morbidity and Mortality of Coronary Artery Disease: A Meta-analysis of Prospective Studies. Heart Lung Circ. 2015 Dec;24(12):1180-90. doi: 10.1016/j.hlc.2015.08.005.

Irwin MR, Olmstead R, Carroll JE. Sleep Disturbance, Sleep Duration, and Inflammation: A Systematic Review and Meta-Analysis of Cohort Studies and Experimental Sleep Deprivation. Biol Psychiatry. 2016 Jul 1;80(1):40-52. doi: 10.1016/j.biopsych.2015.05.014.

Li Y, Zhang X, Winkelman JW, Redline S, Hu FB, Stampfer M, Ma J, Gao X. Association between insomnia symptoms and mortality: a prospective study of U.S. men. Circulation. 2014 Feb 18;129(7):737-46. doi: 10.1161/CIRCULATIONAHA.113.004500. Epub 2013 Nov 13.

Sofi F, Cesari F, Casini A, Macchi C, Abbate R, Gensini GF. Insomnia and risk of cardiovascular disease: a meta-analysis. Eur J Prev Cardiol. 2014 Jan;21(1):57-64. doi: 10.1177/2047487312460020. Epub 2012 Aug 31. Review.

Reviewer #2: This is a large cross sectional study on sleep and mental health. The theme is interesting and may attract readers.

Not only sleep duration, but sleep quality may be important.

I have some minor comments.

1) Please add numbers in tables 2-6, otherwise readers can not tell the numbers are large enough for the analysis.

2) Global scores of PSQI are calculated from scores including sleep duration. You are analyzing sleep quality (defined from PSQI global score) and sleep duration (a part of PSQI). Please discuss this with some reasoning.

Reviewer #3: Comments to the Authors

This paper examined the independent and combined associations of sleep duration and sleep quality with lifetime or 12-month experience of common physical and mental disorders. Sleep duration and sleep quality were assessed using the Pittsburgh Sleep Quality Index while lifetime or 12-month medical and psychiatric diagnoses were established using the WHO Composite International Diagnostic Interview 3.0. Results from this multi-ethnic population-based cross-sectional survey showed that across both 12-month and lifetime diagnoses, sleep duration and sleep quality were independently associated with chronic pain, obsessive compulsive disorder and any mental disorder while sleep quality was additionally associated with major depressive disorder, bipolar disorder, generalized anxiety disorder and any physical disorder. Poor sleep combined with short sleep (< 6hrs/day vs 7-8hrs/day) was associated with the highest number of comorbidities among other sleep combinations. Authors conclude that their findings suggest sleep quality to be a more important indicator for psychological and overall health compared to sleep duration.

I think the paper is detailed, thoughtfully written, well presented and has a well justified underlying rationale. The various sections of the paper from the introduction, methods, statistical analysis, results and discussions were well articulated, easy to read and understand. In general, the results and conclusion appear quite straightforward. However, various clarifications and/or issues that need to be addressed are reported below.

In the introduction, authors state “Secondly, Bin also highlighted that sleep duration and

quality have been conceptualized so distinctly that many have failed to recognise that they are measures of the same underlying phenomenon [3]”….authors should also add the fact that sleep duration and quality measure different constructs.

In the introduction, authors state “……………….association of the two sleep measures with both physical and mental health [4-8], mental health was mainly evaluated only at symptomatic level.” Please authors should clarify what they mean by “symptomatic level”

Can the authors rephrase the use of the word “Aforementioned” in the sentence “Aforementioned, sleep characteristics in general populations have been studied;” Rephrase to something like “As noted previously”

The use of the word “outcome” to refer to sleep measures is confusing. Sleep measures suffices especially since these measures are your independent variables

For mental disorders authors note that “We examined both the lifetime and 12-month prevalence to explore possible……..” Can the authors clarify how they assessed for and determined lifetime or 12-month prevalence?

Statistical analysis

Authors state that “For independent associations of each sleep variable with the various health conditions, sleep quality (poor vs good) and sleep duration (<6h vs 7-8h vs >9h/day) were entered as dependent variables in separate logistic regressions with the other sleep variable being controlled and entered as independent variable.” This statement needs to be clarified. Other sleep variables were being assessed as independent predictors of sleep quality and sleep duration. Otherwise, the model cannot have sleep measures as dependent variables when the examination is between the sleep variables (as independent) and health conditions (as outcome or dependent variable).

“For the combined sleep variable, a single ‘sleep duration & sleep quality’ status with six levels (= sleep, 7- 6h & good 8h/day & good sleep; i.e., reference category; 7-8h/day & =9h & good sleep, =6h & poor sleep, poor sleep, and ) was entered as dependent variable in the multinomial =9h & poor sleep logistic regressions”.

For the above statement, please also clarify what you did by entering your sleep measures or variables as dependent variables.

Authors state, “The use of CAPIs eliminated the chance of a random missing data, except refused or “Not applicable” responses which were minimal”. Minimal is a relative word. Can the authors please clarify what they mean by minimal. For example, how many responses were listed as “Not Applicable (NA)” and was there any sensitivity analyses done that led to the determination of the effect of the NAs?

Authors state, “A complete case analysis was therefore adopted in the current study. Statistical significance was set at p<0.05 level using two-sided tests.” I think it is important to correct for multiple comparisons in order to control for type I error, given the number of statistical analyses performed using the same dataset and variables. Given that the main aim of this study is to investigate the relationship between the independent and combined associations of sleep duration and sleep quality with lifetime or 12-month experience of common physical and mental disorders, you can argue that you only need to correct for your p-values considering the analyses with physical (9 outcomes) and mental disorders (6 outcomes) (p=0.0033 i.e., 0.05/15 analyses). Otherwise, with 12 months and lifetime outcomes separately accounted for, then your family wise error (alpha FWE) would need to be further controlled for.

Table S1 and Table 1 should be configured into 1 table. The whole population should be described by stratified by sleep duration and sleep quality.

Can the authors define what “Any physical disorder” and “Any mental disorder” means?

Table 3 & 5: can the authors please spell out the acronyms on the table in a footnote

Discussion

Can the authors be more explicit when they state “…..some of the challenges associated with studying clinical populations have been circumvented thus providing new insights.” What challenges? I only see author refer to one in the next paragraph. Are there others?

Authors discuss the limitations of the study however they fail to discuss the possible implications on their results. For example, both sleep quality and sleep duration were self-reported, how would these have led to possible misclassifications and possibly affect the effect estimate? Will it drive it towards or away from the null?

6. PLOS authors have the option to publish the peer review history of their article (what does this mean?). If published, this will include your full peer review and any attached files.

Reviewer #1: No

Reviewer #2: No

Reviewer #3: No

---

## [Author Response · Author response to Decision Letter 0]

9 Jun 2020

Claudio Andaloro

Academic Editor

PLOS ONE

Dear editor,

Thank you for your time in reviewing this manuscript. We would also like to thank the reviewer for their constructive comments and I have addressed their comments point-by-point below. We have ensured that our manuscript meets PLOS ONE's style requirements, including those for file naming and updated our Data Availability statement in the cover letter.

Reviewer #1: The background is largely and purposely limited.

These are uncited meta-analyses (sic) that -in different way- assess the relation between sleep duration and quality and health outcomes.

da Silva AA, de Mello RG, Schaan CW, Fuchs FD, Redline S, Fuchs SC. Sleep duration and mortality in the elderly: a systematic review with meta-analysis. BMJ Open. 2016 Feb 17;6(2):e008119. doi: 10.1136/bmjopen-2015-008119.

Gallicchio L, Kalesan B. Sleep duration and mortality: a systematic review and meta-analysis. J Sleep Res. 2009 Jun;18(2):148-58. doi: 10.1111/j.1365-2869.2008.00732.x.

Shen X, Wu Y, Zhang D. Nighttime sleep duration, 24-hour sleep duration and risk of all-cause mortality among adults: a meta-analysis of prospective cohort studies. Sci Rep. 2016 Feb 22;6:21480. doi: 10.1038/srep21480.

Kawada T1. Total sleep time and all cancer mortality: a meta-analysis. Sleep Med. 2020 Apr;68:96. doi: 10.1016/j.sleep.2019.12.029. Epub 2020 Jan 10.

Ge L1, Guyatt G2, Tian J3, Pan B4, Chang Y2, Chen Y4, Li H4, Zhang J4, Li Y4, Ling J3, Yang K5. Insomnia and risk of mortality from all-cause, cardiovascular disease, and cancer: Systematic review and meta-analysis of prospective cohort studies. Sleep Med Rev. 2019 Dec;48:101215. doi: 10.1016/j.smrv.2019.101215.

Kwok CS1, Kontopantelis E2, Kuligowski G3, Gray M3, Muhyaldeen A4, Gale CP5, Peat GM6, Cleator J7, Chew-Graham C6, Loke YK8, Mamas MA1. Self-Reported Sleep Duration and Quality and Cardiovascular Disease and Mortality: A Dose-Response Meta-Analysis. J Am Heart Assoc. 2018 Aug 7;7(15):e008552. doi: 10.1161/JAHA.118.008552.

Yin J et al. J Relationship of Sleep Duration With All-Cause Mortality and Cardiovascular Events: A Systematic Review and Dose-Response Meta-Analysis of Prospective Cohort Studies. Am Heart Assoc. (2017)

Jike M et al. Long sleep duration and health outcomes: A systematic review, meta-analysis and meta-regression. Sleep Med Rev. (2018)

Li Y, Cai S, Ling Y, Mi S, Fan C, Zhong Y, Shen Q. Association between total sleep time and all cancer mortality: non-linear dose-response meta-analysis of cohort studies.

Ma QQ, Yao Q, Lin L, Chen GC, Yu JB. Sleep duration and total cancer mortality: a meta-analysis of prospective studies. Sleep Med. 2016 Nov - Dec;27-28:39-44. doi: 10.1016/j.sleep.2016.06.036.

Stone CR, Haig TR, Fiest KM, McNeil J, Brenner DR, Friedenreich CM. The association between sleep duration and cancer-specific mortality: a systematic review and meta-analysis. Sleep Med. 2019 Aug;60:211-218. doi: 10.1016/j.sleep.2019.03.026.

Lovato N, Lack L. Insomnia and mortality: A meta-analysis. Cancer Causes Control. 2019 May;30(5):501-525. doi: 10.1007/s10552-019-01156-4. Epub 2019 Mar 22.

García-Perdomo HA, Zapata-Copete J, Rojas-Cerón CA. Sleep duration and risk of all-cause mortality: a systematic review and meta-analysis. Sleep Med Rev. 2019 Feb;43:71-83. doi: 10.1016/j.smrv.2018.10.004.

Krittanawong C, Tunhasiriwet A, Wang Z, Zhang H, Farrell AM, Chirapongsathorn S, Sun T, Kitai T, Argulian E. Association between short and long sleep durations and cardiovascular outcomes: a systematic review and meta-analysis. Epidemiol Psychiatr Sci. 2019 Oct;28(5):578-588. doi: 10.1017/S2045796018000379.

Jike M, Itani O, Watanabe N, Buysse DJ, Kaneita Y. Long sleep duration and health outcomes: A systematic review, meta-analysis and meta-regression. Eur Heart J Acute Cardiovasc Care. 2019 Dec;8(8):762-770. doi: 10.1177/2048872617741733.

Itani O, Jike M, Watanabe N, Kaneita Y. Short sleep duration and health outcomes: a systematic review, meta-analysis, and meta-regression. Sleep Med Rev. 2018 Jun;39:25-36. doi: 10.1016/j.smrv.2017.06.011.

Li W, Wang D, Cao S, Yin X, Gong Y, Gan Y, Zhou Y, Lu Z. Sleep duration and risk of stroke events and stroke mortality: A systematic review and meta-analysis of prospective cohort studies. Int J Cardiol. 2016 Nov 15;223:870-876. doi: 10.1016/j.ijcard.2016.08.302.

Liu TZ, Xu C, Rota M, Cai H, Zhang C, Shi MJ, Yuan RX, Weng H, Meng XY, Kwong JS, Sun X. Sleep duration and risk of all-cause mortality: A flexible, non-linear, meta-regression of 40 prospective cohort studies. Sleep Med Rev. 2017 Apr;32:28-36. doi: 10.1016/j.smrv.2016.02.005.

Yang X, Chen H, Li S, Pan L, Jia C. Association of Sleep Duration with the Morbidity and Mortality of Coronary Artery Disease: A Meta-analysis of Prospective Studies. Heart Lung Circ. 2015 Dec;24(12):1180-90. doi: 10.1016/j.hlc.2015.08.005.

Irwin MR, Olmstead R, Carroll JE. Sleep Disturbance, Sleep Duration, and Inflammation: A Systematic Review and Meta-Analysis of Cohort Studies and Experimental Sleep Deprivation. Biol Psychiatry. 2016 Jul 1;80(1):40-52. doi: 10.1016/j.biopsych.2015.05.014.

Li Y, Zhang X, Winkelman JW, Redline S, Hu FB, Stampfer M, Ma J, Gao X. Association between insomnia symptoms and mortality: a prospective study of U.S. men. Circulation. 2014 Feb 18;129(7):737-46. doi: 10.1161/CIRCULATIONAHA.113.004500. Epub 2013 Nov 13.

Sofi F, Cesari F, Casini A, Macchi C, Abbate R, Gensini GF. Insomnia and risk of cardiovascular disease: a meta-analysis. Eur J Prev Cardiol. 2014 Jan;21(1):57-64. doi: 10.1177/2047487312460020. Epub 2012 Aug 31. Review.

We thank the reviewer for the references provided. Majority of the above meta-analyses focused on the relationship between sleep duration and cardiovascular diseases and cancers, as well as with all-cause mortality. We have cited those conducted among cohort studies to complement our introduction. We have also cited those that looked at the relationship between insomnia symptoms and health outcomes in a separate line. 

Reviewer #2: This is a large cross sectional study on sleep and mental health. The theme is interesting and may attract readers. Not only sleep duration, but sleep quality may be important. I have some minor comments.

1) Please add numbers in tables 2-6, otherwise readers cannot tell the numbers are large enough for the analysis.

We have added the numbers as requested. 

2) Global scores of PSQI are calculated from scores including sleep duration. You are analyzing sleep quality (defined from PSQI global score) and sleep duration (a part of PSQI). Please discuss this with some reasoning.

We have discussed this in our methodology under the section of ‘sleep measures’.

Reviewer #3: This paper examined the independent and combined associations of sleep duration and sleep quality with lifetime or 12-month experience of common physical and mental disorders. Sleep duration and sleep quality were assessed using the Pittsburgh Sleep Quality Index while lifetime or 12-month medical and psychiatric diagnoses were established using the WHO Composite International Diagnostic Interview 3.0. Results from this multi-ethnic population-based cross-sectional survey showed that across both 12-month and lifetime diagnoses, sleep duration and sleep quality were independently associated with chronic pain, obsessive compulsive disorder and any mental disorder while sleep quality was additionally associated with major depressive disorder, bipolar disorder, generalized anxiety disorder and any physical disorder. Poor sleep combined with short sleep (< 6hrs/day vs 7-8hrs/day) was associated with the highest number of comorbidities among other sleep combinations. Authors conclude that their findings suggest sleep quality to be a more important indicator for psychological and overall health compared to sleep duration.

I think the paper is detailed, thoughtfully written, well presented and has a well justified underlying rationale. The various sections of the paper from the introduction, methods, statistical analysis, results and discussions were well articulated, easy to read and understand. In general, the results and conclusion appear quite straightforward. However, various clarifications and/or issues that need to be addressed are reported below.

In the introduction, authors state “Secondly, Bin also highlighted that sleep duration and

quality have been conceptualized so distinctly that many have failed to recognise that they are measures of the same underlying phenomenon [3]”….authors should also add the fact that sleep duration and quality measure different constructs.

We have included the above suggestion and highlighted qualitative difference between the two sleep constructs in the introduction.

In the introduction, authors state “……………….association of the two sleep measures with both physical and mental health [4-8], mental health was mainly evaluated only at symptomatic level.” Please authors should clarify what they mean by “symptomatic level”

We have provided additional information to clarify what we meant by “symptomatic level” in the introduction.

 Can the authors rephrase the use of the word “Aforementioned” in the sentence “Aforementioned, sleep characteristics in general populations have been studied;” Rephrase to something like “As noted previously”

We have rephrased the word as suggested.

The use of the word “outcome” to refer to sleep measures is confusing. Sleep measures suffices especially since these measures are your independent variables

We have made the necessary changes throughout the manuscript. 

For mental disorders authors note that “We examined both the lifetime and 12-month prevalence to explore possible……..” Can the authors clarify how they assessed for and determined lifetime or 12-month prevalence?

We have included the further clarification in our methodology under “mental disorders” section. 

Statistical analysis

Authors state that “For independent associations of each sleep variable with the various health conditions, sleep quality (poor vs good) and sleep duration (<6h vs 7-8h vs >9h/day) were entered as dependent variables in separate logistic regressions with the other sleep variable being controlled and entered as independent variable.” This statement needs to be clarified. Other sleep variables were being assessed as independent predictors of sleep quality and sleep duration. Otherwise, the model cannot have sleep measures as dependent variables when the examination is between the sleep variables (as independent) and health conditions (as outcome or dependent variable).

“For the combined sleep variable, a single ‘sleep duration & sleep quality’ status with six levels (= sleep, 7- 6h & good 8h/day & good sleep; i.e., reference category; 7-8h/day & =9h & good sleep, =6h & poor sleep, poor sleep, and ) was entered as dependent variable in the multinomial =9h & poor sleep logistic regressions”. 

For the above statement, please also clarify what you did by entering your sleep measures or variables as dependent variables.

We will address all points above in the following. We will like to clarify that the current study is in fact exploring the association between sleep variables and health conditions, but not sleep variables as predictors of health outcomes or vice versa. The current study is cross-sectional in design and we are therefore unable to establish any casual relationships between sleep and health variables. We had cited these as our limitations. As such, we have included the sleep variable of interest as the dependent and entered all physical disorders + the other sleep variable as independent variables in a single regression model such that for example, the reported odds ratio and CI reflects the relationship of each specific disorder (e.g., hypertension) with sleep quality while controlling for all other physical disorders, any mental disorder and sleep duration. This will also allow us to address problems of comorbidities among participants. We have amended the text and provided an example in our statistical analysis to better clarify this. 

Such statistical method where sleep variable is entered as the dependent variable has been used by other population-based cross-sectional studies examining the relationship between sleep measures and health-related conditions (see Ref 13, 14 and 17).

Authors state, “The use of CAPIs eliminated the chance of a random missing data, except refused or “Not applicable” responses which were minimal”. Minimal is a relative word. Can the authors please clarify what they mean by minimal. For example, how many responses were listed as “Not Applicable (NA)” and was there any sensitivity analyses done that led to the determination of the effect of the NAs?

We have removed the above statement as we deemed it to be inappropriate now. We did not conduct any sensitivity analysis as non-responses appeared to come from various variables such as sociodemographic, health conditions, and sleep variables. 

Authors state, “A complete case analysis was therefore adopted in the current study. Statistical significance was set at p<0.05 level using two-sided tests.” I think it is important to correct for multiple comparisons in order to control for type I error, given the number of statistical analyses performed using the same dataset and variables. Given that the main aim of this study is to investigate the relationship between the independent and combined associations of sleep duration and sleep quality with lifetime or 12-month experience of common physical and mental disorders, you can argue that you only need to correct for your p-values considering the analyses with physical (9 outcomes) and mental disorders (6 outcomes) (p=0.0033 i.e., 0.05/15 analyses). Otherwise, with 12 months and lifetime outcomes separately accounted for, then your family wise error (alpha FWE) would need to be further controlled for.

As noted previously, we have entered all listed physical disorders as independent variables in one model and all mental disorders as independent variables in a separate regression model to look at the association of each specific disorder with the sleep variable of interest, which was entered as dependent variable. Therefore, only 3 main testings or regression models (for physical disorders, 12-month mental disorders and lifetime mental disorders) were run for each of the sleep dependent variables. For example, Table 4, 5 and 6 represent findings from the three models run for the “quality + duration” sleep variable. We have provided further clarity on this in our statistical analyses. We have also stated that the study is exploratory in nature, and thus did not correct for multiple comparisons. Moreover, the idea of correcting of multiple comparisons is still debatable in the literature (O’Keefe, 2003). 

O'Keefe, D.J. (2003), Colloquy: Should Familywise Alpha Be Adjusted?. Human Communication Research, 29: 431-447. doi:10.1111/j.1468-2958.2003.tb00846.x.

Table S1 and Table 1 should be configured into 1 table. The whole population should be described by stratified by sleep duration and sleep quality. 

We are unable to combine S1 into Table 1 as S1 does not include the entire population but only those with the specific physical and mental disorders. We have, however, stratified the whole population by sleep duration and sleep quality in Table 1 as suggested. 

Can the authors define what “Any physical disorder” and “Any mental disorder” means?

We have defined them in the methodology.

Table 3 & 5: can the authors please spell out the acronyms on the table in a footnote

We have spelt out the acronyms as suggested. 

Discussion

Can the authors be more explicit when they state “…..some of the challenges associated with studying clinical populations have been circumvented thus providing new insights.” What challenges? I only see author refer to one in the next paragraph. Are there others?

The two advantages would be (1) providing comparative data to determine whether sleep quality or sleep duration is more related to public health, and (2) be able to link each of these two sleep components to both physical and mental disorders. As highlighted in the introduction, Bin (2016) addressed two major concerns with regards to the literature looking at the relationship between sleep and health, and hence our study hope to address these concerns. These are implied in the same paragraph. 

Authors discuss the limitations of the study however they fail to discuss the possible implications on their results. For example, both sleep quality and sleep duration were self-reported, how would these have led to possible misclassifications and possibly affect the effect estimate? Will it drive it towards or away from the null?

We have discussed the how self-reported sleep indicators would possibly lead to misclassification and affect effect estimate. However, we are unable to confirm the direction of change. 

We hope that we have adequately addressed the reviewer’s comments. Thank you for allowing us to resubmit this revised manuscript for further consideration.

Sincerely, 

Esmond Seow

Research Division

Institute of Mental Health, Singapore

---

## [Decision Letter · Decision Letter 1]

24 Jun 2020

Independent and combined associations of sleep duration and sleep quality with common physical and mental disorders: Results from a multi-ethnic population-based study

PONE-D-19-31356R1

Dear Dr. Seow,

We’re pleased to inform you that your manuscript has been judged scientifically suitable for publication and will be formally accepted for publication once it meets all outstanding technical requirements.

Kind regards,

Claudio Andaloro

Academic Editor

PLOS ONE

Additional Editor Comments (optional):

Reviewers' comments:

Reviewer's Responses to Questions

**Comments to the Author**

1. If the authors have adequately addressed your comments raised in a previous round of review and you feel that this manuscript is now acceptable for publication, you may indicate that here to bypass the “Comments to the Author” section, enter your conflict of interest statement in the “Confidential to Editor” section, and submit your "Accept" recommendation.

Reviewer #1: All comments have been addressed

Reviewer #2: All comments have been addressed

2. Is the manuscript technically sound, and do the data support the conclusions?

Reviewer #1: Yes

Reviewer #2: Yes

3. Has the statistical analysis been performed appropriately and rigorously? 

Reviewer #1: Yes

Reviewer #2: Yes

4. Have the authors made all data underlying the findings in their manuscript fully available?

Reviewer #1: No

Reviewer #2: Yes

5. Is the manuscript presented in an intelligible fashion and written in standard English?

Reviewer #1: Yes

Reviewer #2: Yes

6. Review Comments to the Author

Reviewer #1: The authors have changed the manuscript concerning the major point I raised.

Although this study and the conclusion are not original and should be considered within tens of similar studies, it has the strenght of being represantative of the whole population

Reviewer #2: (No Response)

7. PLOS authors have the option to publish the peer review history of their article (what does this mean?). If published, this will include your full peer review and any attached files.

Reviewer #1: No

Reviewer #2: Yes: Hiroshi Kadotani

---

## [Editor Report · Acceptance letter]

6 Jul 2020

PONE-D-19-31356R1 

Independent and combined associations of sleep duration and sleep quality with common physical and mental disorders: Results from a multi-ethnic population-based study 

Dear Dr. Seow:

I'm pleased to inform you that your manuscript has been deemed suitable for publication in PLOS ONE. Congratulations! Your manuscript is now with our production department. 

Kind regards, 

on behalf of

Dr. Claudio Andaloro 

Academic Editor

PLOS ONE